# Gut–Kidney–Heart: A Novel Trilogy

**DOI:** 10.3390/biomedicines11113063

**Published:** 2023-11-15

**Authors:** Mario Caldarelli, Laura Franza, Pierluigi Rio, Antonio Gasbarrini, Giovanni Gambassi, Rossella Cianci

**Affiliations:** 1Department of Translational Medicine and Surgery, Fondazione Policlinico Universitario A. Gemelli IRCCS, Catholic University of Rome, 00168 Rome, Italy; mario.caldarelli01@icatt.it (M.C.); pierluigi.rio18@gmail.com (P.R.); antonio.gasbarrini@unicatt.it (A.G.); giovanni.gambassi@unicatt.it (G.G.); 2Emergency Medicine Unit, Fondazione Policlinico Universitario A. Gemelli IRCCS, Catholic University of Rome, 00168 Rome, Italy; cliodnaghfranza@gmail.com

**Keywords:** gut microbiota, gut–kidney–heart axis, immune system, dysbiosis, toxins

## Abstract

The microbiota represents a key factor in determining health and disease. Its role in inflammation and immunological disorders is well known, but it is also involved in several complex conditions, ranging from neurological to psychiatric, from gastrointestinal to cardiovascular diseases. It has recently been hypothesized that the gut microbiota may act as an intermediary in the close interaction between kidneys and the cardiovascular system, leading to the conceptualization of the “gut–kidney–heart” axis. In this narrative review, we will discuss the impact of the gut microbiota on each system while also reviewing the available data regarding the axis itself. We will also describe the role of gut metabolites in this complex interplay, as well as potential therapeutical perspectives.

## 1. Introduction

It is now well established that the gut microbiota (GM) can influence the function of extraintestinal organs, including the kidney, the heart, and the brain [1,2].

One of the ways through which the GM is capable of interacting with distant organs is its ability to modulate the immune system [3]. In particular, the GM is made up of different microorganisms which produce several metabolites and interact with the cells at the gut interface, such as immune cells [4,5]. When the GM is in equilibrium, it maintains a balance between inflammatory and anti-inflammatory signals [6]; however, a microbial imbalance can determine local and systemic consequences.

While the gut microbiota influences human health by modulating organ functions, the contrary is also true. For instance, kidney dysfunction is capable of impacting GM homeostasis: in patients with kidney disorders, it has been observed that gut permeability increases [7], while different renal toxins promote the multiplication of gut pathogenic bacteria [8].

The interaction between the gut and the kidney is complex and also involves the cardiovascular system, thus creating an intricate crosstalk, which is often referred to as the “gut–kidney–heart axis” [9]. While the mechanisms underlying its functions are not yet completely clear, our understanding is improving thanks to different studies that have been carried out in the last few years [10].

The main purpose of our narrative review is to evaluate how the microbiome influences renal diseases, as well as the complex crosstalk involving the cardiovascular system. We will then analyze the possible therapeutic implications and the evidence of their effectiveness in these diseases.

## 2. Gut Microbiota and Immunity

The gut microbiota establishes a complex and highly dynamic ecosystem in the host. It consists of all the different microorganisms colonizing the gastrointestinal (GI) tract, including viruses and several species of archaea, protozoa, and fungi [11], which differ from area to area in the gut, and are integral to determining the health of the host. There are six main bacterial phyla colonizing the gut, and Bacteroidetes and Firmicutes make up about 90% of microbiota [12].

The colonization of the GI tract occurs at birth and is influenced by various factors: it has been observed, for instance, that antibiotic therapies administered to children under the age of three are capable of modulating GM significantly with and lasting effects [13], and even the type of delivery can have a severe impact on the composition of the GM [14]. Age in general also plays a role in GM composition: for instance, the GM of a 70 year old is different from that of an infant, and is characterized primarily by a reduced population of Bifidobacteria and an increased amount of Clostridia [15]. Ethnicity has also been linked to differences in microbiota composition [16], which could explain differences in the incidence of diseases in certain populations, and may even prove effective in terms of disease prevention and treatment [17]. While the aforementioned factors are difficult or impossible to change, one of the main drivers of microbiota composition is represented by diet [18], which can instead be modulated to some extent. The impact of diet on microbiota composition has been the object of many different studies, and researchers agree that high fat and high calorie diets have the most negative impact on GM composition [19]. Indeed, a certain grade of variety in the composition of the GM is considered to be normal, yet some composition patterns are associated with worse health outcomes. In these cases, the GM is in a state of disequilibrium, which is commonly referred to as dysbiosis [20], and is characterized, in particular, by a reduction in microbial diversity and an increased presence of some bacterial species.

The term dysbiosis refers to an imbalance of GM in favor of pathogenic microorganisms (defined as pathobionts), followed by a broad spectrum of systemic disorders [21].

A healthy GM is important given all the different roles it can play in the organism, which can be roughly classified as metabolic, mechanical, and immunological (summarized in Table 1).

The GM is responsible for digesting indigestible plant polysaccharides and synthesizing a variety of vitamins and necessary amino acids [22]. In the case of dysbiosis, the way nutrients are absorbed is affected, and this can lead to malnutrition [23] or obesity [24]. GM also plays a role in the metabolism of drugs [25]: it has been observed, for instance, that *Enterococcus faecalis* and *Lactobacillus* spp. can metabolize L-dopa, reducing the efficacy of the drug in patients with Parkinson’s disease [26]. Another field in which the interaction between GM and drugs is being studied is oncology. The importance of GM in cancer is well known [5], but it now appears clear that it can be a key agent in determining the responses to therapies, particularly immunotherapies. For instance, *Bifidobacterium longum, Enterococcus faecium,* and *Bacillus thetaiotaomicron* have appeared to positively impact the response to therapy with PD-1 inhibitors, while other bacteria, such as *Escherichia coli*, have instead been linked to worse responses. Overall, the use of antibiotics before starting therapy has been linked to negative outcomes and reduced progression-free survival [27].

The GM acts as a barrier on the apical surface of the intestinal epithelium, preventing harmful bacteria from colonizing the gut and upregulating the gene expression of mucin, which can limit intestinal microbial translocation [28]. The translocation of components of the gut microbiota, and in particular of lipopolysaccharides (LPS) [29], can lead to an inflammatory reaction, which can be either localized or systemic. While some GM metabolites can drive inflammatory responses, other products provide beneficial effects, as is the case with short-chain fatty acids (SCFAs). SCFAs are produced from fiber fermentation and are the main energy source of the microbiota, but are also key modulators of immunity [30]: SCFAs interact with the immune system through the family of G protein-coupled receptors (GPCRs), specifically through free fatty acid receptors (FFARs) [31]. They also are able to influence phagocytosis and production of reactive oxygen species (ROS), and help to maintain the integrity of the gut barrier [32]. Interestingly, several diseases have been linked to intestinal permeability disruption associated with dysbiosis, ranging from Alzheimer’s disease [33] to other critical conditions [34]. The common denominator in all these conditions seems to be chronic low-grade inflammation [35].

Through its barrier function, the GM plays a role in shaping the immune system of the host [35]; the microbial products of the GM are, indeed, one of the primary sources of the microbe-associated molecular patterns (MAMPs) that bind pattern recognition receptors (PRRs) to innate cells, such as macrophages and natural killers.

As for the innate components of the immune system, the GM plays a key role in shaping the mucosa-associated lymphoid tissue (MALT) of the gut (GALT); the role of the GALT is to identify possible pathogens in a non-specific fashion [36]. In germ-free animal models, it has been observed that the lack of GM reflects the composition of the GALT, which appears to be underdeveloped and incapable of identifying possible pathogens [37]. The interaction between the GM and GALT is regulated through lymphoid tissue inducer cells, which require the stimuli offered by the GM to effectively promote the different components of the GALT [38]. The influence of GM on lymphoid tissue inducer cells occurs through the interaction between PRRs and pathogen-associated molecular patterns (PAMPs) [39]. In particular, Toll-like receptors (TLRs) are among the most important PRRs, and, in the development of a healthy GALT, TLR-2 appears to play a key role [40]. Macrophages and dendritic cells are also present in the GALT, and they take part, as antigen-presenting cells (APCs), in the PAMPs–PRRs interaction. Interestingly, the interaction between the GM and these cells also impacts their differentiation and development; it has been observed, for instance, that a healthy GM is necessary in order for an adequate expression of C-C chemokine receptor type 2 (CCR-2) to be induced, which is needed to guarantee the adequate homing and development of phagocytes [41]. In addition, it has been observed that microbial products such as SCFAs also impact their functionality: SCFAs appear to be capable of inhibiting histone deacetylases (HDACs) while activating G-protein coupled receptors (GPCRs). In germ-free animal models, it has been observed that interferons (IFNs); interleukin (IL)-6, IL-12, and IL-18; and tumor necrosis factor (TNF) are not correctly produced by phagocytes residing in the gut due to epigenetic alterations modulated by SCFAs [42]. Another type of immune cell in the gut which is highly influenced by the GM consists of innate lymphoid cells (ILCs) (e.g., natural killers, NKs). This group of cells presents intermediate characteristics between innate and adaptive immune cells, and it has been observed that PAMPs expressed by the GM are able to influence their differentiation and functionality, particularly in the case of helper-like ILCs [43,44].

**Table 1 biomedicines-11-03063-t001:** Roles of a healthy GM and its impact on immunity.

Roles of the Gut Microbiota	References
Digestion of indigestible plant polysaccharides	[22]
Synthesis of vitamins and necessary amino acids	[22]
Metabolism of drugs	[25]
Barrier on the apical surface of the intestinal epithelium against pathogens	[28]
Immune modulation: production of SCFAs, source of MAMPs, constitution of the GALT, induction of CCR-2 necessary for phagocytosis, differentiation and function of ILCs and lymphocytes	[29,31,35,40,42,43,44]

Abbreviations: SCFAs, short-chain fatty acids; MAMPs, microbe-associated molecular patterns; GALT, gut-associated lymphoid tissue; CCR-2, C-C chemokine receptor type 2; ILCs, innate lymphoid cells.

While ILCs share some characteristics with lymphocytes, they are not part of the adaptive immune system. In the gut, the adaptive immune system is represented mainly by T-lymphocytes and immune-globulin (Ig)A secreting B-lymphocytes [45]. Differentiation of the latter is promoted by different GM metabolites and PAMPs, particularly those called super-antigen-like molecules [46,47], but it is also influenced by the interaction with other cells belonging to the innate and adaptive immune systems. T-lymphocytes are particularly relevant in promoting the adequate differentiation of B-cells, and are themselves influenced by the GM [46]. In particular, it appears that specific bacterial populations are able to promote the differentiation of lymphocytes towards specific subtypes: for instance, colonization by *Clostridium* spp. has been linked to a more abundant population of IL10^+^ Tregs, while segmented filamentous bacteria promote the proliferation of Th17 lymphocytes [48]. SCFAs also play a role in the differentiation of T-lymphocytes, promoting the equilibrium between effector T-cells and T-helpers [49]. It has also been observed that in dysbiotic conditions, the GM tends to promote the expression of Th17 lymphocytes, which have been involved in the pathogenesis of different inflammatory conditions [50].

However, defining what a truly healthy GM is still appears to be an open challenge, given the dynamic equilibrium in which the GM exists. Low variability and the presence of pathogens, however, are considered to be dysbiotic conditions. In such a state, the GM is not able to perform its various activities, thus negatively affecting one’s health status [51]. The human microbiome is becoming a crucial factor in personalized medicine, providing intriguing solutions for a wide range of environmental and metabolic disorders [52].

The variability of microorganisms among individuals and across different spatial and temporal situations requires multidimensional approaches to determine the potential uses of microbiome-based therapeutics [53].

A well-balanced gut microbiome is characterized by newly elucidated connections, such as the gut–brain, gut–liver, and gut–lung axes. These findings have led the gut to be considered as a pivotal organ for overall human health [54].

Nevertheless, the investigation of microbiota predominantly takes into consideration only the bacterial component, leaving out the roles of fungi, viruses, phages, and other microorganisms [55].

Furthermore, a new scenario is linked to the study of the microbial presence of the bacterial microbiome in healthy blood [56]. Identifying potential systemic biomarkers may be beneficial in diagnosing various diseases. Analyzing the blood microbiome represents a noninvasive method for exploring disease biomarkers, potentially enhancing the precision of disease classification and the effectiveness of treatments. However, several unanswered questions persist, including unraveling the intricate relationship between the microbiome and the immune system in both homeostasis and disease.

## 3. The Gut Hypothesis in Cardiovascular Disease

The interaction between the GM and cardiovascular system is bidirectional and is influenced by several factors: in fact, in recent years, there has been talk regarding a “gut hypothesis” (Table 2).

Heart failure (HF) is “a clinical syndrome with symptoms and/or signs caused by a structural and/or functional cardiac abnormality and corroborated by elevated natriuretic peptide levels and/or objective evidence of pulmonary or systemic congestion” [57].

This situation leads to insufficiency of intestinal vascularization with wall edema and mucosal damage, as well as increased intestinal permeability (IP) and intestinal dysbiosis, a condition recognized as “leaky gut” [58,59]. Increased IP and dysbiosis favor the translocation of bacteria and their metabolites into systemic blood circulation, inducing chronic inflammation [59].

Inflammation is one of the main actors in these diseases [60], but diet can also directly impair GM eubiosis: increases in *Bacteroides* spp., *Alistipes* spp., and *Bilophila* spp. have been identified in people following a Western-type diet, which also reduces beneficial bacteria, such as *Lactobacillus* spp., *Roseburia* spp., and *Eubacterium rectale*. Overall, the GM shows a loss of microbial diversity, in turn promoting the development of a highly inflammatory intestinal niche, with a loss of Th17 lymphocytes and increased translocation of bacterial products such as LPS and trimethylamine N-oxide (TMAO) in the bloodstream, which allow inflammation to become systemic [61,62].

Moreover, bacterial translocation leads to the activation of APC, resulting in the production of pro-inflammatory cytokines (TNF-alpha, IL-1, and IL-6) which cause vascular dysfunction and hyperinflammatory conditions [63].

From a molecular perspective, IL-1β, IL-6, TNF-α, matrix metalloproteinase (MMP)-2, MMP-9, TLRs, intercellular adhesion molecule (ICAM)-1, vascular cell adhesion molecule (VCAM)-1, CCR2, and chemokine ligand 2 (CCL2) are all involved in the development of HF [64].

Intestinal permeability also determines alterations in the homeostasis of water and salts [65]. The reabsorption of water and sodium chloride (NaCl) is regulated by various sodium cotransporters and solute carriers, such as Na^+^/H^+^-exchangers (NHE)2, NHE3, and NHE8 [66]. At the enterocyte level, NHE3 is the most expressed apical Na^+^/H^+^ exchanger. Intestinal hypoperfusion results in an increased expression of NHE3 with consequential water and sodium reabsorption, as well as cardiac work overload. Moreover, the expulsion of hydrogen acidifies the intestinal lumen, with a consequent increase in dysbiosis [67].

Specifically, HF determines changes in the composition of GM, with a loss of variety and an increase in pathogenic microorganisms such as *Campylobacter* spp., *Salmonella* spp., *Shigella* spp., *Yersinia enterocolitica* [68], *Candida* spp., and *Chlamydia pneumoniae* [69].

Dysbiosis and intestinal barrier dysfunction could, thus, influence the development of HF by producing metabolites which function as hormones. Phenylacetyl glutamine (PAGln), a phenylalanine metabolite, has been associated with the development of heart failure (both in preserved and reduced ejection fraction) [70]. PAGln also promotes an increase in N-terminal pro-B-type natriuretic peptide (NT-proBNP) values, and appears to have a negative inotropic effect [71].

Interestingly, the degree of severity of HF, measured using the New York Heart Association (NYHA) functional classes, appears to correlate with an increase in the growth of pathogenic gut bacteria: Pasini et al. [67] examined a population comprising 10 healthy controls, 30 patients with NYHA I to II, and 30 patients with NYHA III to IV. The entire population of HF subjects had higher amounts of *Candida, Campylobacter, Shigella*, and *Yersinia* compared to the healthy controls. Overall, within the HF population, the increase in these microbial species was proportional to the severity of the NYHA score. Also, IP was normal in healthy people, while in HF patients, IP was increased by 78.3%.

While the clinical severity of HF does impact the GM composition, the ejection fraction does not seem to play an important role in the modulation of the microbiota. HF is classified, based on left ventricle ejection fraction (LVEF), in HF with a preserved ejection fraction (LVEF ≥ 50%), a mid-range ejection fraction (LVEF = 41–49%) or a reduced ejection fraction (LVEF ≤ 40%) [72]. Hayashi et al. observed that the differences between the microbiomes of patients with HFrEF versus HFpEF were not statistically significant [70].

However, the crosstalk between GM and the heart is not limited to the HF condition alone; it has been associated with the development of different cardiovascular disorders such as atherosclerosis, hypertension, myocardial fibrosis, myocardial infarction, and coronary artery disease [73,74].

Increases in *Lactobacillus* and decreases in *Roseburia* have been associated with atherosclerosis. This situation increases TMAO levels, which are linked to plaque instability and major adverse cardiac events [75,76,77].

Marques et al. demonstrated that a diet rich in fiber modifies the intestinal microbiota and has a protective role in the development of arterial hypertension [78]. They treated mice with deoxycorticosterone acetate, which determines an excess of mineralocorticoids and, therefore, an increase in systolic and diastolic blood pressure. These mice, after consuming a diet rich in fiber, reported changes in the microbiota, with an increase in metabolites such as short-chain fatty acid acetate. Consequently, a significant reduction in blood pressure values was observed.

GM is also linked to coronary heart disease (CHD). Liu et al. documented that, in subjects with CHD, there is an overgrowth of *Lactobacilli* and *Firmicutes* and a decrease in *Bifidobacterium* and *Prevotella* [79].

**Table 2 biomedicines-11-03063-t002:** The “gut hypothesis” in cardiovascular diseases.

Gut and Systemic Alterations	Cardiovascular Consequences	References
Dysbiosis; intestinal permeability Bacterial translocation; microbial metabolites Pro-inflammatory cytokines; chronic inflammation	Heart failure	[59,64,66]
	Homeostasis of water and salts	[64,65]
	Atherosclerosis	[72,73]
	Myocardial fibrosis	[72,73]
	Coronary heart disease	[74,75,78]
	Hypertension	[77]
	Arrythmias	[79]

The intestinal microbiome is related to the development of arrythmias. Gut microbiota-derived metabolites, such as TMAO, LPS, choline, and indoxyl sulphate, promote NOD-, LRR-, and pyrin domain-containing protein 3 (NLRP3) inflammasome activation. This determines the activation of the caspase-1, which generates IL-1b, promoting cardiac inflammation and fibrosis. These changes can affect the correct conduction of the cardiac electrical impulse [80].

## 4. Gut–Kidney Axis, a Bidirectional Talk

The gut–kidney axis represents a bidirectional interaction between the kidney and the GM [81] (Table 3).

On one hand, gastrointestinal diseases could be complications of chronic kidney disease (CKD); on the other hand, gastrointestinal disorders can promote the progression of CKD [82]. GM is involved in the development and progression of metabolic disorders [83] and hyperuricemia [84]; at the same time, uric acid itself and uremic toxins, such as indoxyl sulfate and p-cresyl sulfate, are involved not only in tubulointerstitial fibrosis, glomerular sclerosis, and kidney damage, but also in the disruption of GM balance, thus worsening the intestinal uric acid metabolism [85].

In CKD, urea is accumulated, causing uremic dysbiosis [86]. Dysbiosis can influence the normal functions of the GM, including its ability to maintain the structural integrity of the gut epithelium. Dysbiosis results in a loss of tight junction proteins of the intestinal cells and reduced mucus production, both associated with intestinal barrier dysfunction [87]. In this context, lipopolysaccharide (LPS), a constituent of Gram-negative bacteria and other microbial metabolites, may translocate into systemic circulation [69]. LPS can be recognized by TLR-4 and trigger signaling through myeloid differentiation primary response 88 (MyD88), activating nuclear factor kappa-light-chain-enhancers of activated B cells (NF-kBs) and mitogen-activated protein (MAP) kinases, promoting kidney inflammation [88]. In particular, Watanabe et al. demonstrated the importance of MyD88 signaling in kidney injuries: MyD88 expressed by intestinal epithelial cells causes an increase in the levels of IL-1b, IL-12p40, and IL-17A, all pro-inflammatory cytokines [89]. This generalized inflammatory state is related to acute kidney injury (AKI), CKD, hypertension, nephrolithiasis, and Berger’s disease [90].

In CKD, the gut microbiota shows an increased amount of urease bacteria that are able to metabolize urea to ammonium and ammonium hydroxide, with consequent disruption of the epithelial tight junctions and overall disruption of the intestinal barrier [91], higher luminal pH, impaired function of the intestinal smooth muscle [87], and intestinal motility with shorter transit times and bacterial overgrowth [92]. Moreover, the microbiota of patients with hyperuricemia possesses some characteristics of dysbiosis, particularly in terms of the lack of GM diversity [93] and reductions in Lactobacilli, Bifidobacteria, *Clostridium acidurici 9a,* and *Saccharomices cerevisiae*, which are all capable of degrading uric acid [94]. Moreover, it is worth highlighting that Lactobacilli are SCFA producers and can act in the metabolism of purines. In rats who present with hyperuricemia, for instance, reductions in the amounts of *Roseburia* and *Coprococcus* have been linked to a reduction in SCFA production, with impaired intestinal permeability [95]. In CKD, on the other hand, researchers have observed increased levels of *Escherichia coli* and other pathogens, e.g., Enterobacteriaceae, which are able to produce toxins, such as indoles, p-cresol, and trimethylamine (TMA). Other microorganisms, i.e., Proteobacteria, are involved in kidney failure are through the conversion of nitrogen to ammonia [96].

Overall, studies have shown that during CKD progression, intestinal dysbiosis becomes more severe: the predominant phylum present in this group of patients was Firmicutes, and there was a progressive and drastic reduction in microbial diversity [97]. Therefore, the stage of kidney disease is related to changes in the composition of the microbiome [98]: during AKI, for instance, an increase in *Escherichia* spp. and *Enterobacter* spp. and a decrease in *Lactobacillus*, Ruminococcaceae, *Faecalibacterium*, and Lachnospiraceae were observed. Yang et al. also observed that depletion of the microbiota using combinations of antibiotics has a statistically significant protective effect against the progression of kidney disease [99].

In the early stages of CKD, reductions in GM species diversity have been observed, with an increased presence of the Ruminococcus genus [100]. As renal disease progresses, Bifidobacteria increase while Lactobacilli decrease [101]. Finally, in end-stage renal disease (ESRD), Chang et al. observed a prevalence of the Veillonellaceae, Lactobacillaceae, and Enterobacteriaceae families, as well as a reduced presence of Eubacteriaceae [102].

**Table 3 biomedicines-11-03063-t003:** The gut–kidney interaction.

Gut and Systemic Alterations	Kidney Disease	Microbial Changes	References
Uremic dysbiosis Leaky gut Hyperuricemia Uremic toxins Inflammation	CKD	Increased urease bacteria Increased pathogens (e.g., Enterobacteriaceae) producing uremic toxins Reduction in GM species diversity Increased Ruminococcus genus Increased Bifidobacteria Decreased Lactobacilli Prevalence of Veillonellaceae and Enterobacteriaceae and reduced presence of Eubacteriaceae (ESRD)	[90,92,95,99,100,101]
	AKI	Increase in *Escherichia* spp. and *Enterobacter* spp. Decrease in *Lactobacillus*, Ruminococcaceae, *Faecalibacterium*, and Lachnospiraceae	[89,97]

Abbreviations: CKD, chronic kidney disease; GM, gut microbiota; ESRD, end-stage renal disease; AKI, acute kidney injury.

Moreover, the increase in IP determines the activation of hepatocytes and Kupffer cells, which promotes the release of proinflammatory cytokines, oxidative species, and profibrogenic factors, in turn promoting liver damage and, possibly, non-alcoholic fatty liver disease (NAFLD) [103]. The pathophysiological mechanism by which NALFD causes the progression of renal disease is not yet fully understood, but the insulin resistance observed in NAFLD increases renal damage through the activation of the NF-kB and C-Jun-N-terminal kinase (JNK) pathways [104]. NAFLD also influences the renin-angiotensin system (RAS), which plays a role in the development of CKD [105]. In particular, the activation of angiotensin-converting enzymes (ACEs) and of angiotensin II type 1 receptors (AT1Rs) promotes vasoconstriction and tubular hypertension, reactive oxygen species (ROS) generation, tubulointerstitial inflammation, and fibrosis through the activation of monocyte chemotactic protein-1 (MCP-1) and transforming growth factor (TGF)-β [106,107].

Numerous uremic toxins are produced by the catabolism of proteins introduced by the diet [9]. Some intestinal bacteria metabolize aromatic amino acids into precursors of uremic toxins. There are different pathways involved in the production of uremic toxin precursors starting from aromatic amino acids; for instance, Bacteroidaceae, Bifidobacteriaceae, Clostridiaceae, Enterobacteriaceae, Enterococcaceae, Eubacteriaceae, Fusobacteriaceae, Lachnospiraceae, Lactobacillaceae, Porphyromonadaceae, Staphylococcaceae, Ruminococcaceae, and Veillonellaceae metabolize Tyrosine and Phenylalanine in p-Cresyl sulfate (pCS) [108]. pCS stimulates the production of markers of endothelial damage [109] and determines oxidative stress by stimulating the production of ROS by cardiomyocytes [110].

The metabolism of tryptophan by the intestinal bacteria generates, instead, indole acetic acid (IAA) and indoxyl sulfate (IxS) [9], which appear to be correlated with the progression of renal fibrosis [111].

TMA is obtained through the degradation of phosphatidylcholine/choline, L-carnitine betaine, dimethylglycine, and ergothioneine [112]. TMA enters the bloodstream and reaches the liver, where it is oxidized to TMAO. In murine models, TMAO increases the levels of proinflammatory cytokines, particularly TNF and IL-1β, while also reducing the levels of IL-10 [113]. In in vitro models, it has also been observed that it may be associated with endothelial dysfunction, reduced endothelial self-reparation, and increased adhesion of monocytes through the activation of the protein kinase C (PKC)/NF-κB/VCAM-1 pathways [114].

## 5. Gut, Kidney, and Heart: A Vicious Circle

The gut microbiota can influence human health by modulating organ function. A disequilibrium in gut microbiota composition, called dysbiosis, leads to a “leaky gut”, resulting in systemic bacterial translocation with consequent activation of pro-inflammatory immune pathways. There is close and intricate traffic between the gut, heart, and kidneys; in this relationship, each participant makes a fundamental contribution to maintaining a subtle and delicate balance, but the real protagonist in the onset and progression of organ dysfunction seems to be dysbiosis, causing the subsequent production of harmful metabolites (Figure 1).

GM and its associated metabolites have been implicated in the progression of many cardiovascular diseases (CVDs), such as hypertensive heart disease, atherosclerosis, myocardial infarction, heart failure, and arrhythmias [115].

Dysbiosis has been identified to play a role in enhancing intestinal permeability, enabling microorganisms to move from the gut into the bloodstream. This process results in a state of persistent inflammation. On the other hand, decreased intestinal tissue perfusion, which includes inadequate microcirculation due to heart failure, can determine changes in gut function and permeability, increasing the translocation of microbes and thereby establishing a vicious circle.

Moreover, patients with chronic heart failure (HF) present a reduction in the abundance of beneficial bacterial communities and an increase in the presence of pathogenic bacteria, including *Shigella, Campylobacter, Salmonella*, and *Candida* spp.

A nationwide study in the United States has also documented an elevated occurrence of pathogenic bacteria, such as *Clostridium difficile*, in the fecal samples of individuals with chronic HF, with significantly higher mortality rates in hospital settings among patients with HF [116].

Furthermore, a study utilizing 16S ribosomal DNA (rDNA) analysis revealed that 22 hospitalized patients with HF exhibited reduced levels of bacteria responsible for producing SCFAs, including *Eubacterium rectale* and *Dorea longicatena* [117]. Similarly, a decline in the population of bacteria producing butyrate was also noted in the intestinal flora of individuals with chronic HF. Butyrate plays an anti-inflammatory role in the gut mucosa by enhancing the production of regulatory T cells [118]. Additionally, patients with chronic HF exhibited a significant increase in the expression of microbial genes associated with bacterial LPS biosynthesis and TMAO generation [118].

TMAO is the most extensively studied microbiota biomarker, demonstrating a correlation with the functional class of HF and with B-type natriuretic peptide levels [119].

LPS exhibits increased levels in decompensated HF [120]. These lipopolysaccharides play a pivotal role in various physiological processes, including gut barrier function, inflammation, cardiac contractility, insulin resistance, and endothelial function.

PAGln and phenylacetylglycine (PAGly), both metabolites originating from the gut microbiota, exert their effects through G-protein-coupled receptors and are implicated in platelet function and thrombosis. Thus, their involvement contributes to the development of cardiovascular disease [121]. The presence of these metabolites in blood samples is associated with elevated reactive oxygen production and apoptosis, reduced cell viability, myocardial contraction, and an increased incidence of thrombotic events.

Furthermore, with the proliferation of pathobionts, there is an increase in the production of TMAO and bacteria producing uremic toxins [122].

Li et al., in their meta-analysis on the role of TMAO, focused on highlighting how its concentration in the blood is linearly correlated to mortality in patients with CDK [122]. The main toxic compounds responsible for accelerating the progression of CKD include TMAO, indoxyl sulfate (IS), and p-cresyl sulfate (p-CS) [123]. The existing literature indicates that these metabolites are associated with the onset of renal fibrosis, endothelial dysfunction, and a decline in the glomerular filtration rate (GFR) [124,125].

Moreover, various studies have demonstrated that these compounds are responsible for cardiovascular complications and contribute to heightened morbidity and mortality in individuals with CKD [126]. Furthermore, dysbiosis appears to play a role in the progression from AKI to CKD. In a state of health, a larger population of obligate anaerobe bacteria in the host’s gut helps to maintain the oxygen balance by generating short-chain fatty acids, particularly butyrate. This microbial community facilitates the hydroxylation of the transcription factor hypoxia-inducible factor (HIF) into HIFα, contributing to the adaptation of epithelial cells to physiological hypoxic conditions. Due to disease, an increased presence of facultative anaerobes in the gut impairs the maintenance of oxygen homeostasis, hindering the production of crucial short-chain fatty acids. This leads to prolonged hypoxia, resulting in damage to the gut epithelium. Consequently, pathogens may damage the proximal renal tubule through an inflammatory pathway represented by higher amounts of Th17 cells, influencing angiogenesis and contributing to the manifestation of AKI symptoms as well as the progression to CKD [127].

## 6. Trimethylamine N-Oxide (TMAO) and Other GM Products

The gut–kidney–heart interaction is modulated, among other factors, by uremic toxins. Uremic toxins are the result of impaired kidney function in the context of kidney disease, and their production is highly dependent on the composition of the GM [128]. To be considered an uremic toxin, a compound needs to meet specific criteria; in particular, it needs to be chemically identified and characterized and its levels should directly correlate to the uremic symptoms [129]. In 2021, the American Society of Nephrology updated the definition and classification of uremic toxins, dividing them into endogenous (e.g., IL-6, myoglobin, modified albumin) and exogenous (e.g., uric acid, TMAO) groups [130]. In this context, the GM plays a key role in the production of some of the most potentially harmful toxins. The GM is, indeed, responsible for the production of indoxyl sulfate, p-cresyl sulfate, and dimethyline, which can all determine severe toxicity, particularly for the cardiovascular system [131,132]. These metabolites are produced through the bacterial metabolism of different compounds, such as phenylalanine and tryptophane, which are then further processed by the liver [133]. These uremic toxins can exert several different effects, some directly affecting the kidney (e.g., glomerular sclerosis, renal fibrosis) and others affecting systemic health as well as, as stated above, particularly cardiovascular health [133]. Levels of indoxyl sulfate and p-cresyl sulfate, for instance, have been directly linked to an increased cardiovascular risk for patients undergoing dialysis [134]. GM also produces compounds which are not generally considered to be uremic toxins, but are important in modulating systemic inflammation and can promote kidney damage. LPS, for instance, is among the GM products that can impact kidney function and has been observed in murine models; its modulation through different compounds can improve kidney functionality [135]. Specific gut microbiota can also produce antihypertensive peptides through the fermentation of milk, animal sources, and plants. They exert positive effects on inflammation, insulin resistance, metabolic syndrome, and blood pressure [136]. Furthermore, the gut microbiota produces metabolites, such as acetate, which are able to induce renal tubulointerstitial damage by activating G-protein-coupled receptor 43 (GPR43) [137]. On the other hand, butyrate improves renal fibrosis and tubular damage through the same GPR43 pathway [138].

Among different GM products, TMAO has recently emerged as a key player in different disorders [139]. TMAO is a product of the oxidation of TMA, which in turn is produced from the metabolism of choline-containing products, ergothioneine, and L-carnitine. TMA and TMAO can also be found in seafood. The exact metabolic pathways of these compounds are still not completely understood, but they appear to be influenced both by genetic and environmental factors. Diet, in particular, appears to be a driver of TMAO levels: the consumption of vegetables and reduced intake of red meat are associated with lower TMAO levels [140].

The production of TMAO is also highly dependent on GM composition. Gammaproteobacteria, Betaproteobacteria, Firmicutes, and Actinobacteria are all capable of producing TMAO, while Bacteroidetes does not appear to be able to accomplish this [141]. More specifically, choline and carnitine can be converted into TMA by the gut microbiota; TMA is then absorbed in the intestine, delivered to the liver via the portal vein, and then converted to TMAO by hepatic flavin monooxygenase 3 [2,68].

All the effects determined by TMAO seem, possibly, to explain the underlying link between TMAO levels and cardiovascular diseases that has been observed in the general population [142]. Elevated TMAO levels have also been associated with hypercholesterolemia, platelet hyperreactivity, endothelial damage, and myocardial fibrosis [1,116].

Schuett et al. [143], for instance, treated more than 4000 patients with heart failure (HF) for 9.7 years. They reported that, after excluding factors such as body weight, smoking, hypertension, diabetes, and atrial fibrillation, high levels of TMAO were associated with all-cause mortality in patients with HF. The mechanism through which TMAO aggravates HF is likely to damage vascular endothelial function while also affecting mitochondrial metabolism and promoting myocardial fibrosis [114,144]. Endothelial dysfunction in patients with elevated TMAO levels has been observed in various contexts. Randrianarisoa et al. reported that higher TMAO levels can predict the thickness of the carotid intima–media [145], while in a prospective cohort study by Wu et al., plasma TMAO levels were measured in patients with severe carotid artery stenosis (>70%). The evaluation took place within 3 days prior to carotid artery stenting (CAS), and magnetic resonance was utilized 1 to 3 days afterward. An increased TMAO level was associated with an increased risk of new ischemic brain lesions post-CAS [146]. TMAO was also shown to play a role in peripheral artery disease (PAD). According to Senthog et al., elevated TMAO levels are related to an increased risk of death by 2.7 times in patients with this disease [147], and other studies have shown that patients with PAD presenting with TMAO levels > 2.26 μmol/L exhibit a higher risk of cardiovascular death [148]. Endothelial dysfunction promoted by elevated TMAO levels may also play a role in other contexts; in patients with ST-elevation myocardial infarction, elevated TMAO levels indicate a higher SYNTAX score, and this can be used to evaluate the severity of coronary artery disease, as reported by Sheng et al. [149].

Elevated TMAO levels are also present in CKD. In a study by Cañadas-Garre et al., TMAO seemed to be a promising biomarker for this disease [150], and other studies have hypothesized a causative role of the molecule. They have shown that its reduction through microbiota modulation also appears to slow down the progression of the disease [151,152], which could lead to novel therapeutic approaches using both antibiotics and pre/probiotics.

It has been observed that 3, 3-dimethyl-1-butanol (DMB), a structural analog of choline, may act through inhibiting the production of TMA by the microbiome, in turn reducing the levels of TMAO. However, further studies need to be carried out to evaluate its application in a clinical scenario [2].

In Table 4, we provide a brief description of the main GM metabolites and their effects on overall health.

## 7. The Role of GM Modulation in the Gut–Kidney–Heart Axis

The interaction between the gut, the kidney, and the heart is influenced by all three components, and for this reason, it is possible to discuss a gut–kidney–heart axis [9].

The different parts of this axis all influence each other in different ways, through metabolites, vascular alterations, and immune-mediated mechanisms. While all parts of the axis play important roles in regulating its other parts, the GM is known to be able to determine its regulatory action even over a long distance, mostly through immune-modulation and the metabolites it produces [153].

Diet is one of the most important factors in determining GM composition [154] and it can play a role in the modulation of the gut–kidney–heart axis. According to Chen K. et al., a diet high in saturated fat promotes the proliferation of Firmicutes and Proteobacteria, both related to TMAO production, which, as previously discussed, plays a key role in both cardiovascular and kidney diseases [113].

On the other hand, the Mediterranean diet (MeD) can instead have a beneficial effect on GM composition. The MeD has gained a significant amount of attention in recent years, and it has been shown to offer many advantages compared to a more Western diet [155,156]. De Filippis et al., for instance, have shown that the adoption of the MeD promotes the growth of Prevotellaceae, fecal SCFAs, and lower urinary TMAO compared to other Western diets [157].

Moreover, greater amounts of *Lachnospira* and *Prevotella* were observed in a diet based essentially on vegetables. In addition, Sumida et al. reported that a diet focused mainly on vegetables, which minimizes animal protein intake, preserves the integrity of the intestinal barrier, favors the growth of saccharolytic bacteria, and reduces the production of uremic toxins [158]. Insoluble fibers reach the large intestine unchanged, and are subjected to a process of fermentation by Bacteroidetes and Firmicutes with the production of SCFAs that have a trophic role in the intestinal mucosa and regulatory functions in occludin, claudin-1, and Zonula Occludens-1 [159].

Other diets have also been evaluated in terms of promoting a healthy gut–kidney–heart axis: the high-protein diet, for instance, has gained increasing popularity over the last few years, particularly among those with metabolic syndrome and diabetes. It has been evaluated in the context of kidney disease, but unsurprisingly, it was proven to promote dysbiosis and, potentially, kidney damage [160]. A study by Ang et al. [161] described the potentially positive effects of the ketogenic diet on the gut–kidney–heart axis, reducing inflammation and the Th17 population. Nonetheless, even though this form of dietary intervention holds promise for protection of the kidneys, it also raises concerns due to the heightened risk of an increased intake of potassium and phosphorus [162].

Another interesting possibility is modulating the microbiota by supplementing with prebiotics and probiotics. Prebiotics are indigestible molecules that can promote the growth of beneficial microbial organisms, while probiotics are defined as live microorganisms able to restore a healthy flora in order to provide health benefits to the host [163]. For instance, it has been reported that the oral administration of *Lactobacillus plantarum 299v* induced cardioprotective effects in rats [164]. Lam et al. [165] have observed in murine models that the presence of *Lactobacillus plantarum 299v* in the GM is associated with better outcomes after myocardial infarction. Furthermore, Gan et al. demonstrated that supplementation with *Lactobacillus rhamnosus GR-1* and *Lactobacillus plantarum 299v* has a protective action on the clinical history of HF [166]. Costanza et al. conducted a study on HF patients who were administered a probiotic preparation of *Saccharomyces boulardii* (1 g) for 3 months, and observed reductions in their total cholesterol levels, uric acid levels, and left atrial diameter, as well as improvements in their left ventricular ejection fraction, compared to the control group [167].

Dobrek et al. showed that the use of probiotic, prebiotic, or synbiotic preparations in CKD can be considered a valid therapeutic option [168]. In particular, they analyzed two studies in which prebiotics were administered, six in which probiotics were administered, and three involving the administration of synbiotics. The most common result which was observed was a reduction in the concentration of blood uremic toxins. Ranganathan et al. reported that oral treatment with *Streptococcus thermophilus*, *Lactobacillus acidophilus,* and *Bifidobacterium longum* can slow the progression of kidney disease [169]. They conducted a randomized double-blind study in which 1 gel capsule containing a mix of *Lactobacillus acidophilus KB27, Bifidobacterium longum KB31,* and *Streptococcus thermophilus KB19* was administered for 3 months, for a total of 1.5 × 10^10^ CFU. At the end of the three months, the treated group showed significantly lower levels of blood urea nitrogen, creatinine, and uric acid compared to the control group [169].

However, several studies have indicated that an elevated presence of *Lactobacillus* is associated with several conditions, such as cancer, obesity, and type 2 diabetes [170].

An additional way to modulate the microbiome is the use of conventional drugs.

Lubiprostone is a drug used in the treatment of chronic idiopathic constipation [171]. It activates the chloride channels (CLC)2 by increasing the secretion of water into the intestinal lumen. This mechanism would seem to reduce the absorption and half-life of toxins produced in a dysbiosis setting [168]. Oral administration in rats affected by CDK, for instance, decreased the plasma levels of microbiome-derived uremic toxins, attenuating tubulointerstitial damage and renal fibrosis by restoring the Lactobacillaceae and Prevotella families [2]. However, lubipristone is associated with diarrhea and nausea [172].

Numerous studies have demonstrated how the use of antidiabetic drugs can also affect GM [173]. In particular, it has been shown that sodium glucose cotransporter 2 inhibitors (SGLT2i) can reduce uremic toxins, improve vascular dysfunction, and modulate the gut microbiota [173,174,175]. Lee et al. demonstrated that treatment with Dapagliflozin can reduce the Firmicutes:Bacteroidetes ratio and increased the *Akkermansia muciniphila* population [174]. Treatment with Empagliflozin increases SCFA-producing species, such as *Eubacterium, Roseburia,* and *Faecalibacterium* [176]. However, dapagliflozin concurrently elicits potential side effects. *Paraprevotella* has been identified as a noteworthy bacterium implicated in cardiac structural and functional impairment in a rat model of heart failure. *Desulfovibrio*, as a detrimental genus, has a role as a pivotal sulfate-reducing bacterium in the gut microbiota, introducing hydrogen sulfide and lipopolysaccharide into compromised tissues [177]. Another antidiabetic drug that appears to be able to modulate the microbiome is acarbose, an intestinal alpha-glucosidase inhibitor. Acarbose reduces the pH of the intestinal lumen and, consequently, seems to inhibit the deamination of amino acids [168].

Meclofenamic acid, a non-steroidal anti-inflammatory drug, appears to have the same mechanism as acarbose in terms of modulating the microbiome [168].

Another class of drugs that offer obvious potential for GM modulation is antibiotics. Antibiotics have been used to modulate the GM in different disorders, with encouraging results [178]. Their impact on the gut–kidney–heart axis is also being studied and has proven to be beneficial. Using vancomycin in patients with CKD, for instance, has proven to be effective in reducing gut-derived uremic metabolites [2]. Jiang et al. have demonstrated a reduction in the levels of TMAO in rats being treated with antibiotics [179]. Other authors have also observed a reduction in blood pressure in patients with CKD and cardiovascular disease [59].

Fecal microbiota transplantation (FMT) is another approach which is gaining more and more interest. FMT is currently approved for the treatment of recurrent *Clostridium difficile* infection [180], but is being tested in several diseases—for example, to control intestinal inflammation through the secretion of IL-10 [2]. Recent studies on mice have shown that transferring the microbiome from a mouse with CKD to a germ-free mouse induced renal fibrosis and inflammation and increased the production of uremic toxins in healthy rats [173]; in contrast, FMT from a healthy rat to one with CKD significantly improved renal function [174]. Recently, in patients with renal dysfunction, after washed microbiota transplantation (WMT) through the lower gastrointestinal tract, an increase in gut microbiota diversity has been observed, with a subsequent improvement in renal function and urinary excretion of toxic metabolites [181]. Furthermore, FMT increases the population of butyric-acid-producing Lactobacilli [154]. High levels of butyrate promote the integrity of the intestinal barrier and, consequently, reduce bacterial translocation into the bloodstream [182].

Finally, it is necessary to acknowledge that, while many studies offer interesting results in terms of the potential beneficial effects of GM modulation, we still do not fully understand the consequences of these interventions. In particular, different bacterial species have different benefits in different systems, with some even providing benefits in one condition while being detrimental in another [183]

As a whole, a diverse range of strategies aimed at manipulating the gut microbiome has emerged in mainstream medicine, encompassing both highly targeted interventions and whole-ecosystem approaches. Innovative eubiotics, including with alternative antimicrobials (e.g., triclosan) and natural products (e.g., polyphenols and propolis), show promise as a potential approach, but further studies are required in order to validate their efficacy as microbiome modulators [184].

Thus, further studies are necessary in order for us to fully understand the positive and negative effects of GM modulation in the context of CKD.

## 8. Conclusions

The “microbiota revolution” will bring about fascinating changes in the treatment of cardiovascular pathologies. However, there are still many points to be clarified, and these need to be studied further and supported by scientific evidence. One significant challenge is the precise definition of normal microbiota [63]. Additionally, both the composition of gut microbiota and its products are influenced by several factors, such as age, ethnicity, diet, concomitant medications, and disease [185]. Furthermore, there are limitations in the databases available to study the human gut microbiome [120], and the correlation between the taxonomy and function of the microbiome is not well-defined. The concept of “dysbiosis”, for instance, needs clarification. As discussed throughout the paper, dysbiosis is described as a compositional and functional alteration in the microbiota in individuals with diseases compared with healthy subjects [186]. However, since researchers still do not agree on what constitutes a truly healthy microbiome, it is not clear how to define an impaired one [145]. This leads to the question of when it is appropriate to modulate the microbiome and how to do so. Indeed, GM modulation also influences microbial products, which have the potential to significantly impact health status and, in particular, the gut–kidney–heart axis. Among the different metabolites, TMAO has been the object of various studies, both as a disease marker and as a therapeutic target. Of particular interest in such a sense is the possibility of modulating the GM and its metabolites through the diet, which could offer several advantages. First and foremost of these is the ability to avoid using medication in patients who are likely to already be taking it.

## Figures and Tables

**Figure 1 biomedicines-11-03063-f001:**
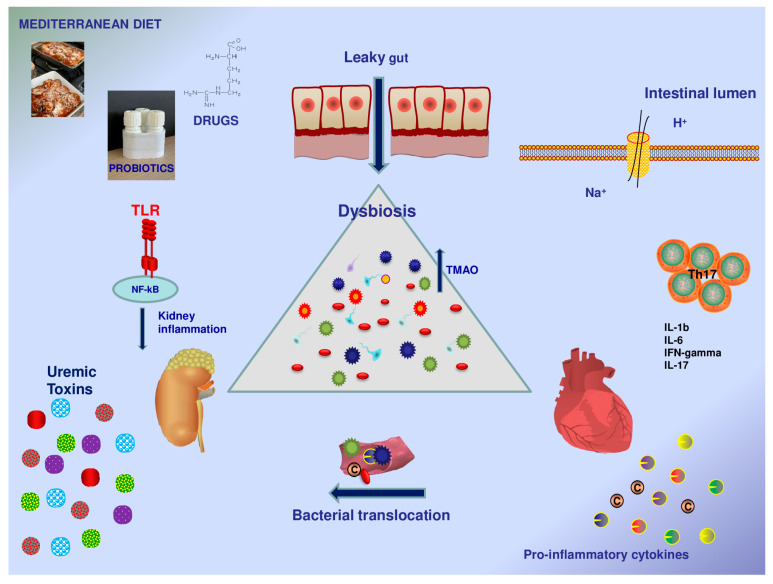
The gut–kidney–heart trilogy. The gut microbiota influences human health by modulating organ function. Disequilibrium in the gut microbiota composition, called dysbiosis, leads to a “leaky gut”, resulting in systemic bacterial translocation with consequent activation of pro-inflammatory immune pathways. Moreover, in kidney disorders, gut permeability increases, while different renal toxins promote the multiplication of gut pathogenic bacteria. The relationship between the gut and the kidneys also involves the cardiovascular system, which has a bidirectional interaction with GM. In this figure, the main pathophysiological mechanisms and external factors that can modulate the gut microbiome, and vice versa, are presented.

**Table 4 biomedicines-11-03063-t004:** Gut microbiota metabolites and their impact on health.

Metabolites	Microbial Origin	Consequences	References
LPS	Component of gut microbiota, constituent of Gram-negative bacteria	Gut and systemic inflammation	[29,82]
SCFAs	Microbial products from fiber fermentation, energy source of gut microbiota	Immune modulation: phagocytosis, ROS production, integrity of the gut barrier, HDAC inhibition and GPCR activation, differentiation of T-lymphocytes	[29,30,31,48]
TMAO	Microbial product deriving from the oxidation of TMA, produced from the metabolism of choline-containing products	Inflammation, plaque instability, MACE, NLRP3 inflammasome activation, cardiac inflammation and fibrosis, kidney damage	[74,106,109,120]
PAGln	Microbial product, phenylalanine metabolite	Heart failure, increase in NT-proBNP, negative inotropic effect	[64,65]
IxS pCS IAA	Microbial products Uremic toxins	NLRP3 inflammasome activation, cardiac inflammation and fibrosis, endothelial damage, kidney damage with tubulointerstitial fibrosis and glomerular sclerosis, uremic dysbiosis	[74,79,103,104]

Abbreviations: LPS, lipopolysaccharide; SCFAs, short-chain fatty acids; ROS, reactive oxygen species; HDACs, histone deacetylases; GPCRs, G-protein coupled receptors; TMAO, trimethylamine N-oxide; TMA, trimethylamine; MACE, major adverse cardiac events; NLRP3, NOD-, LRR- and pyrin domain-containing protein 3; PAGln, phenylacetyl glutamine; NT-proBNP, N-terminal pro-B-type natriuretic peptide; IxS, indoxyl sulfate; pCS, p-cresyl sulfate; IAA, indole acetic acid.

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
