# Peer review of "Gut–Kidney–Heart: A Novel Trilogy"

_biomedicines, 2023, doi:10.3390/biomedicines11113063_

Round 1

Reviewer 1 Report

Comments and Suggestions for Authors

-

Author Response

Rome, 8th November 2023

Dear Editor of “Biomedicines”,

First of all, my coauthors and I would like to thank You sincerely for this opportunity of cooperation, following the submission of the paper “Gut-kidney-heart: A Novel Trilogy” and its possible publication upon “Biomedicines”.

We profoundly thank the reviewers for the comments and useful suggestions aimed at improving the final version of the paper.

We thank You for your constructive critique and we hope the review process has led to an improved manuscript.

If additional changes are warranted, we will make them.

We hope that this revised version of our manuscript may now be found suitable for publication.

Sincerely,

Rossella Cianci, MD, PhD

Reviewer 2 Report

Comments and Suggestions for Authors

Dear authors,

I fond the review Gut-kidney-heart: A Novel Trilogy very ineresting and informing. The manuscript explains how microbiota represents a key factor in determining health and disease in a various patogeny such as neurological, psychiatric, gastrointestinal, cardiovascular  diseases.

One main subject discussed is that gut microbiota may act as an intermediary in the  interaction between kidneys and the cardiovascular system, leading to discussion of  “gut-kidney-heart” axis.

The impact of gut microbiota on each system was presented.

The reached subjects were: Gut microbiota and immunity, The gut hypothesis in cardiovascular disease, Gut-kidney axis, a bidirectional talk, Trimethylamine N-oxide (TMAO), The role of GM modulation in the gut-kidney-heart axis.

Also the role of gut metabolites in this complex interplay and the potential therapeutical perspectives were aproched.

I believe the review is well written, I have though some suggestions, minor completations.

First I suggest in each chapter a table, such Table 1 to summarise the references and some ideeas that come from this references.

I also believe that a table with all acronims uses will raise the value of the manuscript.

Author Response

Rome, 8th November 2023

Dear Editor of “Biomedicines”,

First of all, my coauthors and I would like to thank You sincerely for this opportunity of cooperation, following the submission of the paper “Gut-kidney-heart: A Novel Trilogy” and its possible publication upon “Biomedicines”.

We profoundly thank the reviewer for the comments and useful suggestions aimed at improving the final version of the paper.

This is a point-by-point list of changes made in the paper:

Reviewer 2

Dear authors,

I found the review Gut-kidney-heart: A Novel Trilogy very ineresting and informing. The manuscript explains how microbiota represents a key factor in determining health and disease in a various patogeny such as neurological, psychiatric, gastrointestinal, cardiovascular  diseases.

One main subject discussed is that gut microbiota may act as an intermediary in the  interaction between kidneys and the cardiovascular system, leading to discussion of  “gut-kidney-heart” axis.

The impact of gut microbiota on each system was presented.

The reached subjects were: Gut microbiota and immunity, The gut hypothesis in cardiovascular disease, Gut-kidney axis, a bidirectional talk, Trimethylamine N-oxide (TMAO), The role of GM modulation in the gut-kidney-heart axis.

Also the role of gut metabolites in this complex interplay and the potential therapeutical perspectives were aproched.

I believe the review is well written, I have though some suggestions, minor completations.

First I suggest in each chapter a table, such Table 1 to summarise the references and some ideeas that come from this references.

- Thanks for your comment. We have added tables, as suggested.

I also believe that a table with all acronims uses will raise the value of the manuscript.

- We have added a table with acronyms, as requested.

We thank You for your constructive critique and we hope the review process has led to an improved manuscript.

If additional changes are warranted, we will make them.

We hope that this revised version of our manuscript may now be found suitable for publication.

Sincerely,

Rossella Cianci, MD, PhD

Reviewer 3 Report

Comments and Suggestions for Authors

In this manuscript, authors conducted detailed literature research to support the hypothesis “gut-kidney-heart” axis, which indicates that gut microbiota may act as an intermediary in the close interaction between kidneys and the cardiovascular system. This was a well organized review, and was recommended to be accepted after minor revisions.

1. Please revise ‘spp’ as ‘spp.’ in the species names such as ‘Lactobacillus spp(P2L75)’, ‘Clostridium spp(P3L140)’, ‘Campylobacter spp, Salmonella spp, 185 Shigella spp, ... Candida spp(P4L185)’.

2. Please pay attentions to the superscript format for some phrases, such as ‘Na+/H+-exchangers(P4L179)’ and ‘1.5×1010 CFU(P9L425)’.

3. Figure 1 was a bit blurry.

Comments on the Quality of English Language

Some typo or grammar errors are listed in the comments.

Author Response

Rome, 8th November 2023

Dear Editor of “Biomedicines”,

First of all, my coauthors and I would like to thank You sincerely for this opportunity of cooperation, following the submission of the paper “Gut-kidney-heart: A Novel Trilogy” and its possible publication upon “Biomedicines”.

We profoundly thank the reviewer for the comments and useful suggestions aimed at improving the final version of the paper.

This is a point-by-point list of changes made in the paper:

Reviewer 3

In this manuscript, authors conducted detailed literature research to support the hypothesis “gut-kidney-heart” axis, which indicates that gut microbiota may act as an intermediary in the close interaction between kidneys and the cardiovascular system. This was a well organized review, and was recommended to be accepted after minor revisions.

  1. Please revise ‘spp’ as ‘spp.’ in the species names such as ‘Lactobacillus spp (P2L75)’, ‘Clostridium spp (P3L140)’, ‘Campylobacter spp, Salmonella spp, 185 Shigella spp, ... Candida spp (P4L185)’.

Thank you for your comments. We have revised spp, as suggested.

  1. Please pay attentions to the superscript format for some phrases, such as ‘Na+/H+-exchangers (P4L179)’ and ‘1.5×1010 CFU (P9L425)’.

We have revised superscript format, as suggested.

  1. Figure 1 was a bit blurry.

We have modified the figure 1. Thank you.

We thank You for your constructive critique and we hope the review process has led to an improved manuscript.

If additional changes are warranted, we will make them.

We hope that this revised version of our manuscript may now be found suitable for publication.

Sincerely,

Rossella Cianci, MD, PhD

Reviewer 4 Report

Comments and Suggestions for Authors

In general, the review explores a trending subject with a strong commitment to scientific rigor. The title, "Gut-kidney-heart: A Novel Trilogy," is attention-grabbing and sets the stage for the review. However, upon closer examination, the authors' unique insights and fresh perspectives on this "hot" topic are somewhat unclear. While the manuscript contains a substantial amount of data, there are inconsistencies in the alignment of the text with section titles, and the transitions between topics within sections are not always seamless. I would recommend that the authors address the following points:

1)    The initial structure of the manuscript logically covers sections on the heart-kidney and gut-kidney axes, but the manuscript's focus on the new trilogy, "Gut-kidney-heart," should be more clearly emphasized. It would be advisable to include a dedicated section that elaborates on the interactions within the Gut-kidney-heart axis. Additionally, relocating the figure to this section would enhance overall coherence.

2)    The section “Trimethylamine N-oxide (TMAO)” should be extended to encompass other major uremic toxins, which are integral to the Gut-kidney axis. Furthermore, the table, currently positioned oddly in the conclusions section, should be relocated to this appropriate section.

3)    Microbiome research often produces conflicting findings. The paper would benefit from addressing unclear or unknown data, as this is essential for presenting a balanced view of the topic.

4)    While the manuscript discusses the challenge of defining "dysbiosis," it falls short of providing a concise definition or summarizing the current consensus on this term. Clear definitions are fundamental for maintaining scientific rigor.

5)    The article primarily focuses on the potential benefits of microbiome modulation. However, it's crucial to acknowledge that microbiome research is an ongoing field of exploration, and not all findings and interventions are universally beneficial. Discussing potential downsides is vital for providing a balanced view.

6)    It seems that the knowledge cutoff date for this review is primarily the beginning of 2022, with only a limited number of articles from late 2022 and 2023 being cited. Given the dynamic nature of microbiome research, it's important to recognize the possibility of new findings, studies, and emerging trends in 2023 that may not be covered in this review.

Author Response

Rome, 8th November 2023

Dear Editor of “Biomedicines”,

First of all, my coauthors and I would like to thank You sincerely for this opportunity of cooperation, following the submission of the paper “Gut-kidney-heart: A Novel Trilogy” and its possible publication upon “Biomedicines”.

We profoundly thank the reviewer for the comments and useful suggestions aimed at improving the final version of the paper.

This is a point-by-point list of changes made in the paper:

Reviewer 4

In general, the review explores a trending subject with a strong commitment to scientific rigor. The title, "Gut-kidney-heart: A Novel Trilogy," is attention-grabbing and sets the stage for the review. However, upon closer examination, the authors' unique insights and fresh perspectives on this "hot" topic are somewhat unclear. While the manuscript contains a substantial amount of data, there are inconsistencies in the alignment of the text with section titles, and the transitions between topics within sections are not always seamless. I would recommend that the authors address the following points:

1)    The initial structure of the manuscript logically covers sections on the heart-kidney and gut-kidney axes, but the manuscript's focus on the new trilogy, "Gut-kidney-heart," should be more clearly emphasized. It would be advisable to include a dedicated section that elaborates on the interactions within the Gut-kidney-heart axis. Additionally, relocating the figure to this section would enhance overall coherence.

Thanks for the comment. We have added a paragraph on the trilogy and relocated the figure 1.

2)    The section “Trimethylamine N-oxide (TMAO)” should be extended to encompass other major uremic toxins, which are integral to the Gut-kidney axis. Furthermore, the table, currently positioned oddly in the conclusions section, should be relocated to this appropriate section.

Thanks for the comment. We have added a paragraph on the major uremic toxins and relocated the table.

3)    Microbiome research often produces conflicting findings. The paper would benefit from addressing unclear or unknown data, as this is essential for presenting a balanced view of the topic.

Thanks for the comment. We have discussed some data on microbiome research, as suggested

4)    While the manuscript discusses the challenge of defining "dysbiosis," it falls short of providing a concise definition or summarizing the current consensus on this term. Clear definitions are fundamental for maintaining scientific rigor.

We have defined dysbiosis, as suggested.

5)    The article primarily focuses on the potential benefits of microbiome modulation. However, it's crucial to acknowledge that microbiome research is an ongoing field of exploration, and not all findings and interventions are universally beneficial. Discussing potential downsides is vital for providing a balanced view.

Thanks for the comment. We have discussed potential downsides of microbiota modulation.

6)    It seems that the knowledge cutoff date for this review is primarily the beginning of 2022, with only a limited number of articles from late 2022 and 2023 being cited. Given the dynamic nature of microbiome research, it's important to recognize the possibility of new findings, studies, and emerging trends in 2023 that may not be covered in this review.

We have added some new findings and the emerging trends published in 2023.

We thank You for your constructive critique and we hope the review process has led to an improved manuscript.

If additional changes are warranted, we will make them.

We hope that this revised version of our manuscript may now be found suitable for publication.

Sincerely,

Rossella Cianci, MD, PhD

Round 2

Reviewer 4 Report

Comments and Suggestions for Authors

The paper's quality has been notably enhanced by the authors. While I don't have any major comments on the content of the article, it is crucial to rectify mechanical issues such as the use of Arabic numerals instead of Roman numerals for table numbering, and ensure proper placement of tables after their initial mention in the text before accepting the manuscript.

Comments on the Quality of English Language

The English quality in the manuscript is generally good. However, the paper could benefit from some minor English polishing to enhance its overall quality.

Author Response

Rome, 12th November, 2023 

Dear Editor of “Biomedicines”, 

first, my coauthors and I would like to thank You sincerely for this opportunity of cooperation, following the submission of the paper “Gut-kidney-heart: A Novel Trilogy” and its possible publication in “Biomedicines”. 

We profoundly thank the reviewer for the comments and useful suggestions aimed at improving the final version of the paper. 

REVIEWER 4, round 2

The paper's quality has been notably enhanced by the authors. While I don't have any major comments on the content of the article, it is crucial to rectify mechanical issues such as the use of Arabic numerals instead of Roman numerals for table numbering, and ensure proper placement of tables after their initial mention in the text before accepting the manuscript.

Comments on the Quality of English Language

The English quality in the manuscript is generally good. However, the paper could benefit from some minor English polishing to enhance its overall quality.

Thanks for your suggestion. We have replaced the Roman numerals with Arabic ones (as described in the 'Instructions for authors') and have replaced the tables, as suggested. We have revised typos throughout the text.

We thank You for your constructive critique and we hope the review process has led to an improved manuscript.

We hope that this revised version of our manuscript may now be found suitable for publication.

Sincerely,

Rossella Cianci